# Progressive Point Cloud Denoising with Cross-Stage Cross-Coder Adaptive Edge Graph Convolution Network

## ABSTRACT

Due to the limitation of collection device and unstable scanning process, point cloud data is usually noisy. Those noise deforms the underlying structures of point clouds and inevitably affects downstream tasks such as rendering, reconstruction and analysis. In this paper, we propose a Cross-stage Cross-coder Adaptive Edge Graph Convolution Network ($C^2$AENet) to denoise point clouds. Our network uses multiple stages to progressively and iteratively denoise points. To improve the effectiveness, we add connections between two stages and between the encoder and decoder, leading to the cross-stage cross-coder architecture. Additionally, existing graph-based point cloud learning methods tend to capture local structure. They typically construct a semantic graph based on semantic distance, which may ignore Euclidean neighbors and lead to insufficient geometry perception. Therefore, we introduce a geometric graph and adaptively calculate edge attention based on the local and global structural information of the points. This results in a novel graph convolution module that allows the network to capture richer contextual information and focus on more important parts. Extensive experiments demonstrate that the proposed method is competitive compared with other state-of-the-art methods. The code will be made publicly available.

## CCS CONCEPTS

• **Computing methodologies → Computer vision tasks**.

## KEYWORDS

Point cloud denoising, Graph convolution network

## 1 INTRODUCTION

As 3D sensing technology advances swiftly, point clouds are increasingly becoming a preferred format for representing 3D data, resulting in extensive applications in various 3D vision fields like autonomous driving [5, 20], augmented reality [29, 38] and 3D action recognition [8, 9, 45]. Nonetheless, the acquisition process introduces noise into point clouds due to inherent hardware errors, as well as interference from human, environmental, and other factors. This noise presents substantial challenges for subsequent tasks like surface reconstruction [2, 18] and semantic segmentation [36, 46]. Hence, addressing the issue of point cloud denoising

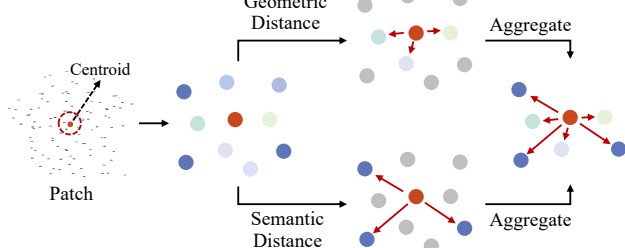

(a) Graph construction in our proposed method

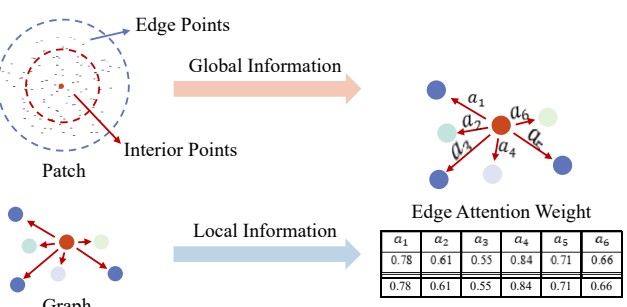

(b) Overview of adaptive edge attention

**Figure 1: (a) Illustration of graph construction in the proposed method. The patch is extracted from the noisy point cloud. Color indicates point semantic distance. Simultaneously considering geometric and semantic distances allows the aggregated graph structure capture more comprehensive contextual information. (b) Illustration of adaptive edge attention. To better learn the structural information of point clouds, we adaptively compute edge attention based on local and global information.**

becomes crucial in ensuring the accuracy and reliability of downstream applications.

Traditional methods for point cloud denoising have demonstrated certain success [7, 13, 14, 17, 21, 23, 31, 40, 44, 47]. However, these approaches often rely on geometric prior knowledge and lack robustness when dealing with point clouds containing high-level noise or non-uniform sampling. A recent trend, fueled by the success of network architectures tailored for point clouds [32], has given rise to deep learning-based denoising methods. Despite their effectiveness, these data-driven methods face challenges in terms of generalization, such as Convolutional Neural Network (CNN). To improve generalization, CNN-based methods in high-level visual tasks typically emphasize multi-stage networks. Similarly, when confronted with point clouds contaminated by unknown high-level noise, these methods tend to iteratively denoise the point clouds

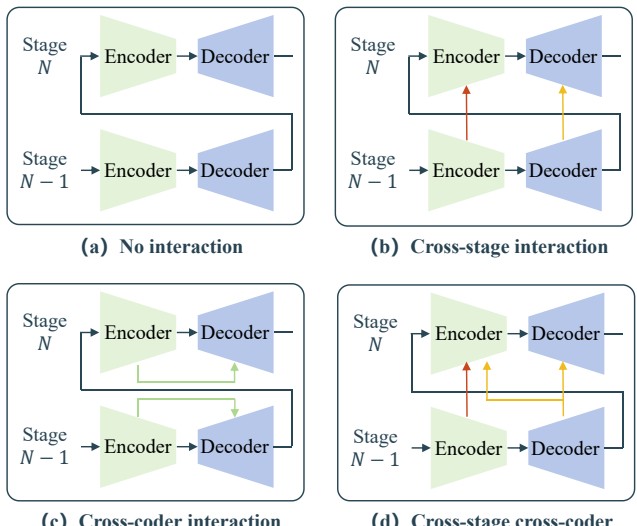

**(a) No interaction**

**(b) Cross-stage interaction**

**(c) Cross-coder interaction**

**(d) Cross-stage cross-coder**

**Figure 2: Illustration of different multi-stage frameworks. (a) Vanilla multi-stage: the decoder output of the previous stage is input to the next stage (black line). (b) Cross-stage interaction: the intermediate results of encoder and decoder flow to the next stage (orange line and yellow line). (c) Cross-coder interaction: the intermediate results of encoder are integrated into decoder (green line). (d) Cross-stage cross-coder interaction: the intermediate results flow across adjacent stages and across encoder and decoder.**

during testing to mitigate generalization issues [4, 19]. Nevertheless, they do not incorporate additional operations during training to account for differences in residual noise across various stages [26, 33, 43]. This oversight can lead to either excessive or insufficient denoising. At present, RePCN-Net [3] addresses this concern by employing a recurrent network architecture for iterative denoising. But it utilizes a single network, limiting its ability to accurately identify noise levels across different stages. By contrast, IterativePFN [6] adopts network stacking, allowing the network to learn denoising at multiple stages with varying noise levels. This approach enhances the model's capability to handle complex noise scenarios in point clouds.

Despite the success of the above methods, a few issues persist. Firstly, they tend to concentrate on learning local structures based on semantic graphs, which results in a deficiency in geometric perception capabilities. To address this limitation, we introduce additional geometric graphs and obtain a graph structure with a larger receptive field. This method can harness the unique advantages of different graph structures to capture more contextual information, as illustrated in Figure 1. On one hand, the importance of different neighbors to the centroid is different in the graph. On the other hand, interior points with more complete neighborhood information have more effective structural information compared to edge points. Therefore, we construct an adaptive edge attention (AEA) module that considers both local and global structural information of points to adaptively learn the weights of the graph structure.

Based on this strategy, we propose an Adaptive Edge Graph Convolution (AEConv) module which provides a more comprehensive model for complex data. Secondly, existing methods often overlook interactions between multiple denoising stages, which may lead to incomplete information transfer or information loss. To tackle this issue, we propose the incorporation of a cross-stage cross-coder architecture, ensuring that the information from the previous stage is fully propagated and utilized in the subsequent stages, as depicted in Figure 2. Based on AEConv module and cross-stage cross-coder architecture, we present a **Cross-stage Cross-coder Adaptive Edge Graph Convolution Network ($C^2$AENet)**, which effectively promotes efficient information flow across different denoising stages while enhancing the performance of graph convolution. The main contributions of this work are as follows:

- We propose a cross-stage cross-coder framework, incorporating a multi-stage information transfer mechanism to ensure that the valuable information of the previous stage is fully utilized.
- We propose an Adaptive Edge Graph Convolution (AEConv) module that leverages local and global structural information to adaptively learn edge attention, effectively capturing the local structures of point clouds.
- Extensive experiments as well as the ablation studies demonstrate the effectiveness and the contribution of each key component involved in the proposed method.

## 2 RELATED WORK

### 2.1 Traditional Denoising Methods

Traditional point cloud denoising can be generally classified into filter-based methods and optimization-based methods.

**Filter-based methods.** Inspired by bilateral filtering, Fleishman et al. [11] used bilateral filtering to denoise 3D mesh models. Digne et al. [7] proposed a bilateral filter for point clouds based on the work of Fleishman et al., taking into account both the geometric and normal distances of points. Zhang et al. [44] proposed a point cloud denoising method based on the principal component analysis (PCA) and bilateral filter. Bilateral filtering-based methods can yield the expected results, but they may exhibit poor denoising effects near sharp edges and require higher computational time. In addition to bilateral filtering, guided filtering is also an effective method. Han et al. [13] adopted the point position as the guidance information and proposed a point position guided filtering method. Subsequently, iterative normal guided filtering [14], anisotropic point set denoising algorithm [40], and point cloud denoising method based on multi-normal strategies [23, 47] have been gradually proposed.

**Optimization-based methods.** Optimization-based denoising methods typically treat the denoising process as an optimization problem constrained by geometric priors. They are mainly divided into Moving Least Squares (MLS)-based and Locally Optimal Projection (LOP)-based. Inspired by the MLS method, Alexa et al. [1] defined a smooth manifold surface at a set of points close to the original surface. Subsequently, MLS-based methods have been gradually proposed, such as Robust MLS (RMLS) [10] and Robust Implicit MLS (RIMLS) [28]. Although MLS-based methods can reconstruct smooth surfaces, they are constrained by the parameters of local surfaces, resulting in a relatively poor description of details in

 

**Figure 3: Framework of the proposed point cloud denoising method. Here, the working principles of the $(N-1)$-th stage and the $N$-th stage are presented. The network structure is the same for each stage, where the encoder consists of three dynamic AEConv modules, and the decoder is composed of multi-layer perceptron (MLP). The color of each point in the point cloud represents its P2M value, ranging from 0 to 100.**

sharp features. To alleviate this problem, Lipman et al. [22] first proposed applying the LOP method to surface approximation of point sets. Afterward, more and more methods have been proposed, such as weighted LOP (WLOP) [16], edge-aware resampling (EAR) [17], feature-preserving LOP operator (FLOP) [21], and continuous weighted LOP (CLOP) [31]. The LOP-based methods have certain generality, but they also have the problem of over-smoothing.

## 2.2 Deep Learning-based Denoising Methods

In recent years, with the success of deep learning in the field of computer vision, deep learning-based methods have been widely applied to point cloud denoising. Yu et al. proposed an Edge-aware Point Set Consolidation Network (ECNet) [41], leveraging edge-aware techniques to facilitate the consolidation of point clouds. PointProNet [34] utilized CNNs to filter noisy 2D height maps and then reproject them into 3D space. Similarly, Li et al. [24] employed 2D height maps to estimate normals, thereby achieving point position updates. Rakotosaona et al. proposed a two-stage denoising network based on PCPNet [12], called PointCleanNet [33], to remove outliers and reduce noise in unordered point clouds. Pistilli et al. [30] proposed a graph convolutional layer based denoising network called GPDNet. The Pointfilter proposed by Zhang et al. [43] used the autoencoder architecture for point cloud denoising. Additionally, Luo et al. [26] employed neural networks to estimate distribution scores and denoised the point cloud through gradient ascent. Mao et al. [27] utilized normalizing flows and noise disentanglement techniques to predict the displacement vectors of noise. Chen et al. [3] proposed a recurrent network architecture called RePCD-Net. To ensure fast convergence of points to the clean surface, IterativePFN [6] modeled the iterative filtering

process within the network. In addition to supervised denoising methods, unsupervised denoising methods also play a crucial role, such as TotalDenoising [15] and DMRDenoise [25].

## 3 PROPOSED METHOD

### 3.1 Overview

In this paper, we construct a $C^2$AENet based on an encoder-decoder architecture to extract rich feature representations for point cloud denoising. The encoder consists of three dynamic AEConv modules, aiming at effectively extracting feature information at different scales. The decoder employs a multi-layer perceptron (MLP), taking multi-scale features as input, to predict noise for input points. This method follows a multi-stage framework. We show two adjacent stages of the framework in Figure 3.

Specifically, the $(N-1)$-th stage and the $N$-th stage respectively learn to predict the residual noise of different stages. To improve effectiveness, we introduce a cross-stage cross-coder architecture to propagate important information learned from the $(N-1)$-th stage to the $N$-th stage. These early-stage information will effectively promote the learning and optimization in the subsequent stage, reducing errors and biases of the model in complex tasks. Finally, the noise predicted at multiple stages is aggregated and apply them to the noisy point cloud for denoising. We detail the adaptive edge graph convolution network and the cross-stage cross-coder architecture below.

### 3.2 Adaptive Edge Graph Convolution Network

We denote the ground-truth point cloud as $\tilde{P} = \{\tilde{p}_i\}_{i=1}^{N} \in \mathbb{R}^{N \times 3}$ and the noise as $U = \{u_i\}_{i=1}^{N} \in \mathbb{R}^{N \times 3}$, where $N$ is the number of

**Figure 4: AEConv mainly consists of a graph construction module, an adaptive edge attention module, and an edge graph convolution module. The graph construction module generates the corresponding graph structure for $N$ input points based on geometric distance and semantic distance. The graph structure includes centroid features and neighbor features, where edge features are obtained by subtracting centroid features from neighbor features. The adaptive edge attention module calculates the edge attention weight based on local edge features and global structural information. Finally, the output feature map is obtained through the edge graph convolution module.**

points. The input noisy point cloud is generated as follows:

$$P = \tilde{P} + U. \tag{1}$$

The goal of point cloud denoising is to predict noise $U$ for recovering the original point cloud $\tilde{P}$.

**Graph Construction Module.** Graph Convolutional Networks (GCN) is able to effectively handle non-Euclidean data and achieves excellent performance on a range of applications[39]. However, conventional GCN typically focuses on learning high-level feature knowledge but neglects the original geometry structure[37]. To improve efficiency, the network denoises multiple points simultaneously, making the construction of the graph structure particularly crucial. To improve this problem, we propose an AEConv module that is able to simultaneously consider the geometric distance and hierarchical distance between points to construct different graphs, and adaptively learns the edge attention weights of the graph structure based on local and global structural information. Therefore, AEConv can learn more comprehensive intrinsic correlations and global correlations of point clouds. Its structure is shown in Figure 4.

We take each point in the noisy patch as a centroid and connect it to its neighbors to construct a graph $\mathcal{G} = (\mathcal{V}, \mathcal{E})$, where $\mathcal{V} = \{1, \ldots, N\}$ is a vertex set and $\mathcal{E} \subseteq \mathcal{V} \times \mathcal{V}$ is the edge set. In implementation, we construct the graph using the $k$-nearest neighbors ($kNN$) of each point without self-loop, and the number of neighbors in each graph is set to 16. We define $p_i$ as the centroid of a graph, and $\mathcal{J}(i) = \{j | (i, j) \in \mathcal{E}\}$ as the set of points in its neighborhood. Based on two different-level features, we capture the geometric and semantic relationships between pairs of points, generating geometric graphs $\mathcal{G}_g$ and semantic graphs $\mathcal{G}_s$ with different neighborhood information,

$$\begin{aligned} \mathcal{G}_g^i &= kNN_{j=1,\cdots,N}(||p_i - p_j||^2), \\ \mathcal{G}_s^i &= kNN_{j=1,\cdots,N}(||f_i - f_j||^2). \end{aligned} \tag{2}$$

where $f_i$ and $f_j$ represent the feature vectors of the $i$-th and $j$-th points and are initialized as the point coordinates $p_i$ and $p_j$, respectively.

**Adaptive Edge Attention Module.** To enhance the representation capability of the graph structure, we construct an AEA module to adaptively learn the edge attention weights of the graph structure. First, we aggregate the different neighborhoods of the two graph structures as follows:

$$\mathcal{G}_f = \mathcal{G}_g \cup \mathcal{G}_s. \tag{3}$$

We define the edge feature $e_{ij} \in \mathbb{R}^{C_{in}}$ as the relative feature vector between the centroid $p_i$ and the neighboring point $p_j$:

$$e_{ij} = f_i - f_j, \tag{4}$$

where $C_{in}$ is the input dimension of the point features. The local edge features help to understand the intrinsic correlations of the graph, and interior points with more complete neighborhood information can capture more effective structural information compared to edge points. Therefore, we simultaneously compute the edge attention for the graph structures of $N$ points, enabling the network to focus on the more crucial points in the global structure and the more significant vertices in the graph. We define the attention weight of an edge in the aggregated graph $\mathcal{G}_f$ as $a_{ij}$, which represents the importance of the edge feature $e_{ij}$ for the current centroid $p_i$. It can be formulated as:

$$a_{ij} = \sigma(\Phi(e_{ij})), \tag{5}$$

where $\Phi$ is a non-linear function used for local channel context aggregation, and $\sigma$ represents the Sigmoid non-linear activation function. The heatmap visualization of edge attention is shown in Figure 5. We separately visualize the edge attention weights of the graph structures for edge points and interior points. As can be seen from the figure, the graph constructed from interior points as centroids obtains larger weights compared to the edge points.

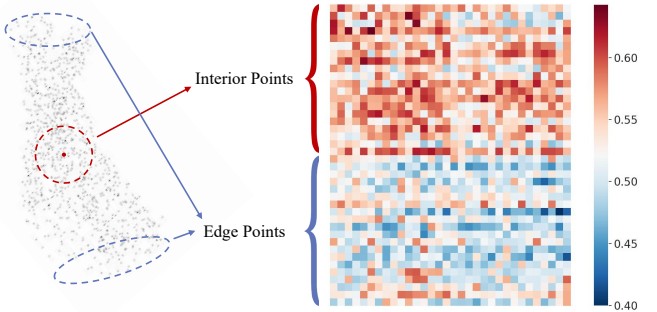

**Figure 5: Given a patch of noisy point cloud (left), the heatmap visualization of edge attention is computed and shown (right). The horizontal axis represents the vertices in the graph, and the vertical axis represents the points in the patch. The color bar provides a mapping from attention weights to the color space. In the heatmap, red color indicates higher edge attention weights, while blue indicates lower weights.**

This further demonstrates that interior points with more complete neighborhood information can capture more effective structural information. Based on this weight, AEConv adaptively learns the importance of neighbors to better capture the graph structural information.

Then, we integrate the weighted edge features $\bar{e}_{ij}$ with centroid features $f_i$ to combine global shape information and local neighborhood information,

$$\bar{e}_{ij} = a_{ij} \cdot e_{ij}, \quad h_{ij} = (f_i \| \bar{e}_{ij}), \tag{6}$$

where $\cdot$ denotes vector dot product, $(\cdot \| \cdot)$ denotes concatenation, and $h_{ij}$ represents weighted graph features.

**Edge Graph Convolution Module.** In the edge graph convolution operation, we update vertex features based on neighborhood information to capture the local features of the weighted graph, which can be formulated as follows:

$$f'_i = g_\Theta^{(l)}(\Sigma_{j:(i,j)\in\mathcal{E}} h_{ij}), \tag{7}$$

where $g_\Theta^{(l)} : \mathbb{R}^{F_l} \times \mathbb{R}^{F_l} \rightarrow \mathbb{R}^{F_{l+1}}$ represents the $l$-th graph convolutional layer, $\Theta$ is the set of learnable parameters, and $F_l$ is the feature dimension at $l$-th layer. To better capture the non-linear relationships between vertices in the edge-fusion graph, we implement $g_\Theta^{(l)}$ by MLPs. Subsequently, we utilize symmetric functions for feature aggregation, capturing local geometric structure while maintaining permutation invariance. Due to the dynamic changes in feature representation, the static graph may become ineffective. Therefore, we dynamically update the graph structure during the learning process.

### 3.3 Cross-stage Cross-coder Architecture

In multi-stage networks, the encoder-decoder architectures of different stages are independent of each other. Despite its success, it overlooks the interactions between multiple stages. This could lead to incomplete information transfer or information loss. To efficiently leverage the valuable information from previous stages, we facilitate information transfer between stages.

Specifically, we construct a cross-stage cross-coder architecture. For the encoder, we first apply linear mapping to the features of the corresponding layers of the encoder and decoder at the current stage. This makes the features more suitable for the learning and optimization of the next stage network. Then cross-coder is achieved by feature addition to supplement the lost context information. Finally, it is connected to the corresponding layer in the encoder of the next stage. For the decoder, we fuse multi-scale features from the decoders of two stages, enabling the model to obtain more comprehensive information. In the end, our architecture achieves extensive information propagation between multi-stage networks, effectively alleviating the issues of incomplete information transfer or information loss.

### 3.4 Training Setup

Due to the significant increase in computational and time complexity caused by large-scale point clouds, we treat it as a local problem. During the training process, the patches randomly obtained through the $kNN$ algorithm are fed into the network. To improve efficiency, denoising is simultaneously applied to all points on the patches. During the testing process, multiple patches are extracted from a point cloud for denoising. Subsequently, the best denoised point cloud is obtained by utilizing a patch stitching mechanism [6] to combine different patches.

The loss function used to optimize the network $\mathcal{L}$ is defined as:

$$\mathcal{L} = \frac{1}{T} \sum_{i=0}^{T} \|\Delta_i - u_i\|_2^2, \tag{8}$$

where $i$ represents the current number of stages, $T$ represents the total number of denoising stages, $\Delta$ represents the predicted noise, and $u$ represents the real noise. Therefore, the final loss is the sum of the losses from different stages of the network.

## 4 EXPERIMENTAL RESULTS

### 4.1 Dataset and Implementation Details

To evaluate the effectiveness of the proposed method, we conduct experiments on the **synthetic PUNet dataset** [42]. The training set consists of 120 point clouds sampled using Poisson disk sampling from 40 meshes, with resolutions of 10K, 30K, and 50K points. Then, Gaussian noises with standard deviations ranging from 0.5% to 2% of the bounding sphere's radius are added to the point clouds. The testing set consists of 40 point clouds sampled using Poisson disk sampling from 20 meshes, with resolutions of 10K and 50K points. For the testing point clouds, we add Gaussian noise with standard deviations of 1%, 2%, and 2.5% of the bounding sphere's radius. In addition to the synthetic datasets, we also conduct experiments on the **real-world Kinect v1 and Kinect v2 datasets** [35] which consists of 73 and 72 real-world scans acquired by Microsoft Kinect v1 and Kinect v2 cameras.

Two commonly-used metrics including the Chamfer Distance (**CD**) and the Point2Mesh distance (**P2M**) are used as the criteria for performance evaluation. Specifically, lower values of **CD** and **P2M** indicate better point cloud denoising performance. During training, the network parameters are optimized using the Adam optimizer

**Table 1: Comparison results on the PUNet dataset. CD is multiplied by $10^5$, P2M is multiplied by $10^5$. The best results are marked in BOLD.**

| Point Cloud Denoising Models | Publication | 10K points | | | | | | 50K points | | | | | |
|---|---|---|---|---|---|---|---|---|---|---|---|---|---|
| | | 1% noise | | 2% noise | | 2.5% noise | | 1% noise | | 2% noise | | 2.5% noise | |
| | | CD↓ | P2M↓ | CD↓ | P2M↓ | CD↓ | P2M↓ | CD↓ | P2M↓ | CD↓ | P2M↓ | CD↓ | P2M↓ |
| PointCleanNet [33] | CGF 2020 | 36.86 | 15.99 | 79.26 | 47.59 | 104.86 | 69.87 | 11.03 | 6.46 | 19.78 | 13.70 | 32.03 | 24.86 |
| GPDNet [30] | ECCV 2020 | 23.10 | 7.14 | 42.84 | 18.55 | 58.37 | 30.66 | 10.49 | 6.35 | 32.88 | 25.03 | 50.85 | 41.34 |
| DMRDenoise [25] | ACM MM 2020 | 47.12 | 21.96 | 50.85 | 25.23 | 52.77 | 26.69 | 12.05 | 7.62 | 14.43 | 9.70 | 16.96 | 11.90 |
| PDFlow [27] | ECCV 2022 | 21.26 | 6.74 | 32.46 | 13.24 | 36.27 | 17.02 | 6.51 | 4.16 | 12.70 | 9.21 | 18.74 | 14.26 |
| ScoreDenoise [26] | ICCV 2021 | 25.22 | 7.54 | 36.83 | 13.80 | 42.32 | 19.04 | 7.16 | 4.00 | 12.89 | 8.33 | 14.45 | 9.58 |
| Pointfilter [43] | TVCG 2020 | 24.61 | 7.30 | 35.34 | 11.55 | 40.99 | 15.05 | 7.58 | 4.32 | 9.07 | 5.07 | 10.99 | 6.29 |
| IterativePFN [6] | CVPR 2023 | 20.56 | 5.01 | 30.43 | 8.45 | 33.52 | 10.45 | 6.05 | 3.02 | 8.03 | 4.36 | 10.15 | 5.88 |
| **$C^2$AENet (Ours)** | - | **19.60** | **4.59** | **30.09** | **8.00** | **32.68** | **9.74** | **5.81** | **2.83** | **7.58** | **4.05** | **9.89** | **5.66** |

for 100 epochs, with a batch size of 4, and an initial learning rate of 1e-4. The learning rate is scaled to 0.5 times when the network parameters are not updated for 10 consecutive epochs.

## 4.2 Performance Comparison on PUNet

We compare our method with state-of-the-art point cloud denoising methods, including PointCleanNet [33], GPDNet [30], DMRDenoise [25], PDFlow [27], ScoreDenoise [26], Pointfilter [43], IterativePFN [6]. The performance comparison results are shown in Table 1. From the results, we can draw the following conclusions:

- Our method consistently outperforms the competing methods across all three noise levels. Notably, while methods such as GPDNet, ScoreDenoise, and Pointfilter show promising results on low noise levels, they face significant challenges when confronted with samples with high noise levels. By contrast, our method exhibit excellent performance on both low and high noise levels. This highlights the stability and effectiveness of our method, making it a more reliable choice for denoising tasks across various noise conditions.
- The performance difference among different methods is more pronounced at low resolution compared to high resolution. Consequently, excelling at both low and high resolutions proves to be challenging. Our method not only performs well at high resolution samples but also demonstrates superiority at low resolution samples.

For a more intuitive comparison, we present the visual results of denoised point clouds generated by different methods, as shown in Figure 6. In general, it can be observed that ScoreDenoise and PDFlow exhibit poor preservation of shape structure in the detailed regions. Additionally, PDFlow shows the weakest performance in global point cloud denoising, marked by extensive blue regions. Our method demonstrates better visual results in terms of denoising and preservation of shape structure. For a more comprehensive evaluation, additional visual and experimental results on the PUNet dataset are included in the supplementary material. Specifically, it includes more visual comparison results on the PUNet dataset, runtime comparisons of different denoising methods, and performance comparisons on higher noise levels.

**Table 2: Comparison results on the Kinect v1 and Kinect v2 datasets. CD is multiplied by $10^5$, P2M is multiplied by $10^5$. The first, second, and third of the two indicators are marked in red, blue and green, respectively.**

| Method | Kinect v1 | | Kinect v2 | |
|---|---|---|---|---|
| | CD↓ | P2M↓ | CD↓ | P2M↓ |
| PointCleanNet [33] | 13.73 | 8.75 | 22.48 | 13.29 |
| GPDNet [30] | 14.83 | 8.69 | 23.09 | 11.78 |
| DMRDenoise [25] | 22.78 | 12.89 | \ | \ |
| PDFlow [27] | 14.01 | 9.14 | 20.83 | 12.13 |
| ScoreDenoise [26] | 13.22 | 8.18 | 19.66 | 11.08 |
| Pointfilter [43] | 13.77 | 7.91 | 18.85 | 10.29 |
| IterativePFN [6] | 13.20 | 8.43 | 18.69 | 10.92 |
| **$C^2$AENet (Ours)** | 13.09 | 8.29 | 18.92 | 10.83 |

## 4.3 Visual Results on Real-world Scanned Data

The noise in real-world scanned data is unknown and complex. In addition to the synthetic PUNet dataset, we also conduct experiments on real-world scanned data from the Kinect v1 and Kinect v2 datasets. The specific results are shown in Table 2. Specifically, our method achieves the best results on the **CD** metric of the Kinect v1 dataset and the top-3 performances on the other metrics. Therefore, our method not only performs best on synthetic datasets, but also proves effective on real-world datasets. For a more comprehensive understanding, the supplementary materials include visualizations of the denoising results on real-world scanned data.

## 4.4 Ablation Study

To demonstrate the effectiveness of the cross-stage cross-coder architecture, AEConv module and multi-stage denoising scheme, we performed the following ablation studies:
**Contribution of the cross-stage cross-coder architecture.** To demonstrate the contribution of our cross-stage cross-coder architecture, we compare the performance obtained by using five

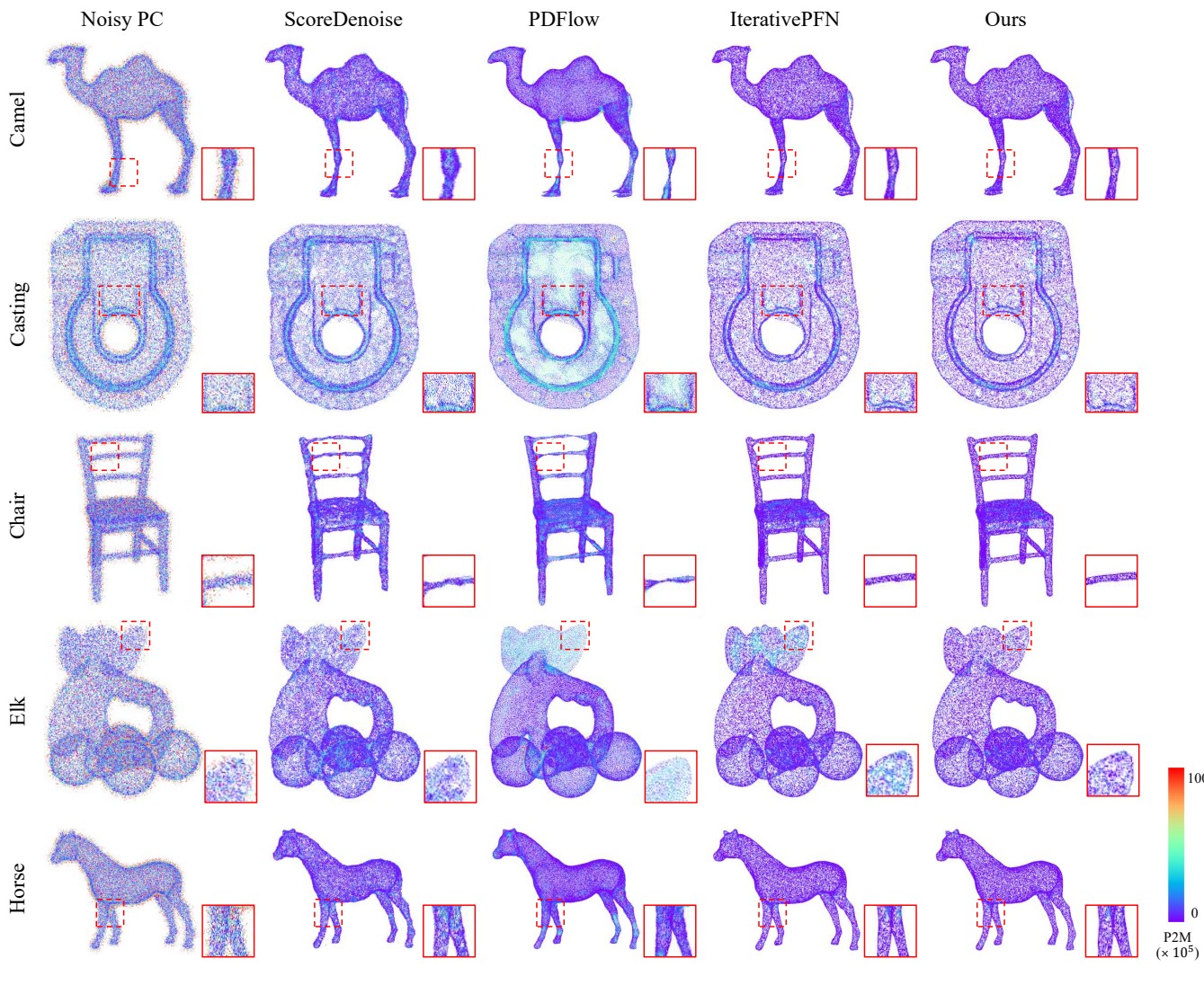

**Figure 6: Visualization of denoised point clouds obtained by four denoising methods, where the color of each point represents its P2M value. Specifically, we visualize the denoising results of Camel, Casting, Chair, Elk, and House selected from the PUNet dataset. The noisy point cloud contains Gaussian noise with standard deviation of 2% of the bounding sphere's radius, and it has a resolution of 50K points. The red box displays the denoising results of different methods in the detailed areas.**

different connection architectures. We consider the 4-stage network without interaction as our baseline. The experimental results are shown in Table 3. From the table, we can observe that adding cross-stage connections of encoder and decoder separately or simultaneously to the baseline model can slightly improve the denoising performance on low noise levels. However, there is no significant improvement observed on high noise levels. On this basis, the linear mapping of cross-stage features integrates the feature information of the previous stage to make the transfer information more suitable for the learning of the next stage. Therefore, this architecture effectively enhances the denoising ability of the network, particularly when addressing high levels of noise.

**Contribution of the AEConv module.** To demonstrate the effectiveness of the AEConv module, we conduct ablation studies. The experiment mainly compare the performance contributions of the EdgeConv [37] baseline, multi-level graph structure, and the AEA module. The results are shown in Table 4. In the table, we use GS to represent multi-level graph structures. From the table, it can be observed that the combination of geometric and semantic information enables the network to capture richer contextual information. The AEA module adaptively learns edge attention based on local and global structural information, which makes the network focus on more important regions. This effectively enhances the learning ability and denoising performance of the network.

**Table 3: Experimental results of five different connection architectures on the PUNet database. CD is multiplied by $10^5$, P2M is multiplied by $10^5$. The best results are indicated in BOLD.**

| Method | 10K points | | | | | | 50K points | | | | | |
|---|---|---|---|---|---|---|---|---|---|---|---|---|
| | 1% noise | | 2% noise | | 2.5% noise | | 1% noise | | 2% noise | | 2.5% noise | |
| | CD↓ | P2M↓ | CD↓ | P2M↓ | CD↓ | P2M↓ | CD↓ | P2M↓ | CD↓ | P2M↓ | CD↓ | P2M↓ |
| Base | 19.70 | 4.83 | 30.40 | 8.37 | 33.77 | 10.51 | 6.05 | 3.02 | 7.98 | 4.31 | 11.01 | 6.44 |
| Base+Cross-Stage (Encoder) | 19.72 | 4.70 | 30.26 | 8.17 | 33.68 | 10.39 | 5.89 | 2.89 | 7.94 | 4.27 | 11.76 | 7.02 |
| Base+Cross-Stage (Decoder) | 19.72 | 4.77 | 30.20 | 8.15 | 33.84 | 10.52 | 5.95 | 2.94 | 7.80 | 4.19 | 10.86 | 6.36 |
| Base+Cross-Stage Cross-Coder | 19.66 | 4.63 | 30.11 | 8.00 | 33.81 | 10.42 | 5.89 | 2.89 | 7.85 | 4.22 | 11.67 | 6.92 |
| Base+Cross-Stage Cross-Coder (FC) | **19.60** | **4.59** | **30.09** | **8.00** | **32.68** | **9.74** | **5.81** | **2.83** | **7.58** | **4.05** | **9.89** | **5.66** |

**Table 4: Ablation results of the AEConv module on the PUNet database. CD is multiplied by $10^5$, P2M is multiplied by $10^5$. The best results are indicated in BOLD.**

| Method | 10K points | | | | | | 50K points | | | | | |
|---|---|---|---|---|---|---|---|---|---|---|---|---|
| | 1% noise | | 2% noise | | 2.5% noise | | 1% noise | | 2% noise | | 2.5% noise | |
| | CD↓ | P2M↓ | CD↓ | P2M↓ | CD↓ | P2M↓ | CD↓ | P2M↓ | CD↓ | P2M↓ | CD↓ | P2M↓ |
| EdgeConv | 21.85 | 4.97 | 30.49 | 8.40 | 33.55 | 10.48 | 5.95 | 2.93 | 7.97 | 4.29 | 10.58 | 6.15 |
| EdgeConv + GS | 21.71 | 4.91 | 30.32 | 8.22 | 33.00 | 10.07 | 5.85 | 2.87 | 7.79 | 4.18 | 10.15 | 5.85 |
| EdgeConv + GS + AEA | **19.60** | **4.59** | **30.09** | **8.00** | **32.68** | **9.74** | **5.81** | **2.83** | **7.58** | **4.05** | **9.89** | **5.66** |

**Table 5: Experimental results of different number of stages on the PUNet database. CD is multiplied by $10^5$, P2M is multiplied by $10^5$. The best results are indicated in BOLD.**

| Method | 10K points | | | | | | 50K points | | | | | |
|---|---|---|---|---|---|---|---|---|---|---|---|---|
| | 1% noise | | 2% noise | | 2.5% noise | | 1% noise | | 2% noise | | 2.5% noise | |
| | CD↓ | P2M↓ | CD↓ | P2M↓ | CD↓ | P2M↓ | CD↓ | P2M↓ | CD↓ | P2M↓ | CD↓ | P2M↓ |
| Our network with 1 stage | 21.72 | 5.42 | 32.74 | 9.90 | 38.84 | 14.09 | 6.35 | 3.21 | 10.12 | 5.79 | 16.00 | 10.27 |
| Our network with 2 stages | 20.71 | 4.86 | 30.96 | 8.60 | 34.25 | 10.76 | 5.91 | 2.90 | 8.63 | 4.71 | 14.73 | 9.23 |
| Our network with 4 stages | 19.60 | **4.59** | **30.09** | **8.00** | **32.68** | **9.74** | **5.81** | **2.83** | **7.58** | **4.05** | **9.89** | **5.66** |
| Our network with 8 stages | **19.15** | 4.62 | 30.31 | 8.18 | 33.73 | 10.73 | 6.05 | 3.02 | 7.96 | 4.22 | 10.78 | 6.17 |

**Comparison of the number of stages.** To substantiate the effectiveness of multi-stage denoising scheme, we conduct comparative experiments on the number of stages. We use $C^2$AENet as the baseline and only vary the number of stages in the network architecture here. The experiment evaluates networks with 1, 2, 4, and 8 stages, and the comparative results are presented in Table 5. The results reveal that the performance gains achieved by the multi-stage denoising scheme tend to be stagnant when the number of stages reaches to 4. At this time, increasing the number of stages does not further improve performance. Consequently, we set the network architecture with 4 stages as default.

To more fully demonstrate the contribution of the key components of our approach, we also add ablation experiments in the supplementary material. They include experimental results of the cross-stage cross-coder architecture, the AEConv module, and the multi-stage denoising scheme on higher noise levels.

## 5 CONCLUSION

In this paper, we present a Cross-stage Cross-coder Adaptive Edge Graph Convolution Network ($C^2$AENet) for point cloud denoising. We introduce a cross-stage cross-coder architecture to alleviate the issues of incomplete information transfer or information loss through efficient information flow across different denoising stages. In addition, it simultaneously utilizes the effective geometric information and semantic information to capture richer contextual information, and employs adaptive edge attention to focus the network on more important parts. Extensive experiments on the PUNet dataset as well as the ablation studies demonstrate the effectiveness and the contribution of each key component involved in the proposed method. Furthermore, the edge attention heatmap and denoising visual results provide a direct explanation for our method.

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
