# OpenReview forum: "Progressive Point Cloud Denoising with Cross-Stage Cross-Coder Adaptive Edge Graph Convolution Network"
_acmmm.org/ACMMM/2024/Conference — MM2024 Poster_

### Official Review · Reviewer_SpTY · 2024-04-30

**Rating:** 3
**Confidence:** 3

**Summary:**

This paper proposes a cross-stage cross-coder adaptive edge graph convolutional network (C2AENet) for denoising point clouds, using multiple stages to iterate step-by-step through the denoised points. To improve efficiency, this paper adds connections between the two stages as well as between the encoder and decoder, resulting in a cross-stage cross-coder architecture. C2AENet computes edge attention adaptively by constructing a geometric graph and based on the local and global structural information of the points. This yields a novel graph convolution module that enables the network to capture richer contextual information and focus on more important parts. Extensive experiments have demonstrated the competitiveness of the proposed method compared to other state-of-the-art methods.

**Strengths:**

The novelty of this paper is incremental, by applying a cross-stage cross-coder adaptive edge graph convolutional network to iterative point cloud denoising, and from the results it makes sufficient improvements to the denoising techniques for point clouds.

The theory of this paper is general but the experiments are detailed. Intuitively, the C2AENet proposed in this paper seems to be a simple component improvement and multi-stage comprehensive analysis of point cloud networks.

The presentation quality of this paper is overall excellent, but parts of the paper still need to be improved to be clearer. See the Limitations for details.

The method proposed in this paper is aimed at denoising point clouds and has some application in real life applications.

**Limitations:**

1. The "analysis" in the abstract is not a downstream task.

2. What is the significance of the repeated elements in the data table in Figure 1? Also, the "Edge Points" and some of the fonts in the figure seem ambiguous or wrong to me.

3. The cross-stage and cross-coder in the paper seem to be similar to the common way of learning hierarchical networks, can the authors explain the difference between the proposed method and them?

4. How do the authors differentiate semantic distances based on point coordinates, are they just taking the supervised conditions and using them as input conditions?

5. From Eqs. 2-7, the edge convolution generation approach proposed in the paper seems to be extremely similar to the EdgeConv proposed for DGCNN.

6. It is suggested to add complexity comparison results or deconstruct the network parameters of each component for this paper's method in the experimental section.

**Suitability:**

2

---

### Official Review · Reviewer_pnjH · 2024-05-11

**Rating:** 5
**Confidence:** 4

**Summary:**

The paper introduces a novel deep learning-based method, C2AENet, for denoising point clouds. The proposed network utilizes a cross-stage cross-coder architecture to progressively refine the denoising process, enhancing information flow across stages. It also introduces an Adaptive Edge Graph Convolution (AEConv) module that leverages both geometric and semantic information to adaptively learn edge attention, allowing the network to focus on structurally significant regions of the point cloud. The method demonstrates competitive performance compared to state-of-the-art techniques and is shown to be effective on both synthetic and real-world datasets.

**Strengths:**

1. Innovative Architecture: The paper presents a new network architecture that effectively handles the progressive denoising of point clouds.
2. Cross-Stage Cross-Coder: The proposed method for information propagation between stages is a significant contribution that improves upon traditional multi-stage approaches.
3. Adaptive Edge Graph Convolution: The AEConv module is a novel approach that captures richer contextual information by adaptively learning edge attention.
4. Extensive Evaluation: The authors provide a thorough experimental evaluation, demonstrating the effectiveness of the method on multiple datasets.
5. State-of-the-Art Performance: The method outperforms existing techniques, indicating its potential for real-world applications.

**Limitations:**

1. Complexity: The cross-stage cross-coder architecture and AEConv module may introduce increased computational complexity.
2. Generalization: The paper primarily focuses on denoising tasks; it is unclear how well these techniques generalize to other point cloud processing tasks.
3. Implementation Details: The paper could benefit from more information on the practical implementation aspects, such as training time and resource requirements.
4. Model Size: The paper does not discuss the size of the final model, which is critical for applications with constrained computational resources.
5. Template Violation: The paper does not adhere to the standard conference template, which may affect the readability and presentation of the research.

**Suitability:**

3

---

### Official Review · Reviewer_vAWt · 2024-05-25

**Rating:** 3
**Confidence:** 3

**Summary:**

This paper focuses on the point cloud denoising task and develops a cross-stage cross-coder adaptive edge graph convolution network to denoise the original point clouds by introducing graph neural networks. The main contributions of this paper involve the multiple denoising stages and geometric graph construction strategy. The former aims to progressively and iteratively denoise points and the latter computes the edge weights based on the structure information of points.

**Strengths:**

1. The proposed multi-stage denoising methods for point cloud dataset is interesting.
2. Empirical experiments are sufficient, especially extensive ablation studies.

**Limitations:**

1. The authors claim one of the contributions of this paper is the geometric graph based the point coordinates. However, the classical DGCNN method seems to use the point coordinates to compute the edge weight. The authors should carefully demonstrate the difference between this work and the previous works.
3. The baseline models used are mostly from 2020 and 2021. The authors should compare them with the baseline models of recent years to verify the effectiveness of the proposed models.
4. There exists a lot of typos, such as P4 N432 “…the points and are initialized as the point coordinates”. The authors should carefully check this paper to ensure that there are no grammar typos.

**Suitability:**

2

---

### Official Review · Reviewer_S7mG · 2024-05-27

**Rating:** 4
**Confidence:** 4

**Summary:**

This paper proposes a method for denoising point clouds, called C2AENet, employs a multi-stage approach to iteratively denoise point clouds. They combine local and global structural information to adaptively calculate edge attention, allowing the network to capture richer contextual information and focus on more important parts of the point cloud. The method demonstrates competitive performance compared to state-of-the-art techniques, showing significant improvements in both synthetic and real-world datasets.

**Strengths:**

1. The use of both geometric and semantic distances in graph construction allows the network to capture a more comprehensive set of contextual information, improving its ability to denoise point clouds. The adaptive edge attention mechanism enables the network to focus on the most crucial parts of the point cloud, improving the overall quality of the denoised output.
2. The experiments are sufficient. The method is evaluated on the PUNet dataset and real-world Kinect datasets, demonstrating its effectiveness across different noise levels and resolutions.

**Limitations:**

1. Figure 1(b) contains incongruous characters that need to be optimized.
2. Directly using features from the previous stage may cause error accumulation, leading the model to fall into a suboptimal solution in the earlier stage and affecting the current stage.
3. Considering the features from two stages simultaneously doubles the model parameters and affects its efficiency. The number of parameters and running time should be compared in the experiment section.
4. In Section 3.4, this paper achieves the final result through patch-level denoising. It is necessary to further analyze whether this approach affects denoising performance, such as the surface consistency between patches.
5. Treating the denoising problem as a local problem neglects global constraints, such as the geometric position and contextual relationships between patches.
6. There are many methods that also consider geometric and semantic distances, and it would be better to cite them：
[1] Jiang et al. DHGCN: Dynamic Hop Graph Convolution Network for Self-supervised Point Cloud Learning. In AAAI 2024.
[2] Wei et al. Wei M, Wei Z, Zhou H, et al. AGConv: Adaptive graph convolution on 3D point clouds. In IEEE TPAMI 2023.
[3] Lin et al. Learning of 3D Graph Convolution Networks for Point Cloud Analysis. In IEEE TPAMI 2022.

**Suitability:**

2

---

### Meta-Review · Area_Chair_u5cF · 2024-06-27

**Recommendation:** Accept (Poster)
**Confidence:** 5

**Metareview:**

All reviewers tend to accept this paper. Authors please carefully address reviewer comments like implementation details, method clarity, experiments etc and include them.